# Developing an Artificial Intelligence-Based Representation of a Virtual Patient Model for Real-Time Diagnosis of Acute Respiratory Distress Syndrome

**DOI:** 10.3390/diagnostics13122098

**Published:** 2023-06-17

**Authors:** Chadi S. Barakat, Konstantin Sharafutdinov, Josefine Busch, Sina Saffaran, Declan G. Bates, Jonathan G. Hardman, Andreas Schuppert, Sigurður Brynjólfsson, Sebastian Fritsch, Morris Riedel

**Affiliations:** 1Jülich Supercomputing Centre, Forschungszentrum Jülich, 52428 Jülich, Germany; 2School of Engineering and Natural Science, University of Iceland, 107 Reykjavik, Iceland; 3SMITH Consortium of the German Medical Informatics Initiative, 07747 Leipzig, Germany; 4Joint Research Centre for Computational Biomedicine, University Hospital RWTH Aachen, 52074 Aachen, Germany; 5School of Engineering, University of Warwick, Coventry CV4 7AL, UK; 6School of Medicine, University of Nottingham, Nottingham NG7 2RD, UK; 7Department of Intensive Care Medicine, University Hospital RWTH Aachen, 52074 Aachen, Germany

**Keywords:** high-performance computing, machine learning, ICU, ARDS, surrogate model, virtual patient

## Abstract

Acute Respiratory Distress Syndrome (ARDS) is a condition that endangers the lives of many Intensive Care Unit patients through gradual reduction of lung function. Due to its heterogeneity, this condition has been difficult to diagnose and treat, although it has been the subject of continuous research, leading to the development of several tools for modeling disease progression on the one hand, and guidelines for diagnosis on the other, mainly the “Berlin Definition”. This paper describes the development of a deep learning-based surrogate model of one such tool for modeling ARDS onset in a virtual patient: the Nottingham Physiology Simulator. The model-development process takes advantage of current machine learning and data-analysis techniques, as well as efficient hyperparameter-tuning methods, within a high-performance computing-enabled data science platform. The lightweight models developed through this process present comparable accuracy to the original simulator (per-parameter R^2^ > 0.90). The experimental process described herein serves as a proof of concept for the rapid development and dissemination of specialised diagnosis support systems based on pre-existing generalised mechanistic models, making use of supercomputing infrastructure for the development and testing processes and supported by open-source software for streamlined implementation in clinical routines.

## 1. Introduction

Respiratory diseases endanger the ability of the respiratory system to supply the body with oxygen and to eliminate carbon dioxide sufficiently, potentially causing life-threatening consequences. These conditions are caused on one hand primarily by damaging the pulmonary tissue through, for instance, infection, toxic effects of inhaled gases or fluids or trauma. On the other hand, the lung can be affected indirectly as a side-effect of diseases of other organs [1]. Early diagnosis and treatment are essential to achieve positive outcomes for patients [2,3,4,5]. Critically ill patients who require treatment in an Intensive Care Unit (ICU) are at high risk of developing respiratory disease, one of the most serious of which is Acute Respiratory Distress Syndrome (ARDS), a condition that was first described by Ashbaugh et al. [6]. ARDS is still the subject of intensive research due to its high incidence in ICU patients as reported by Confalonieri et al. (10.4% of total ICU admissions) and high mortality rate as highlighted by Le et al. (30–55% of affected patients) [3,5].

ARDS is further characterised by its heterogeneity and the difficulty with respect to diagnosing it, leading clinicians and researchers to establish the “Berlin Definition” by which ARDS onset is defined as a ratio of Partial Pressure of Arterial Oxygen (P_a_O_2_) to Fraction of Inspired Oxygen (F_i_O_2_) (P/F ratio) of less than 300 mmHg in combination with bilateral opacities in pulmonary imaging and absence of hypervolemia and heart failure [7]. Furthermore, this definition classifies the severity of the condition to be inversely proportional to the value of the P/F ratio. Despite widespread research activities in this field, which were even intensified during the COVID-19 pandemic, effective treatment methods of ARDS are still lacking, resulting in a high mortality rate [3,5]. In fact, Bellani et al. highlight that ARDS diagnosis is still delayed or missed in two thirds of patients, leading to severe outcomes [8]. The management of ARDS patients, thus, usually remains supportive with lung-protective mechanical ventilation, prone positioning and extracorporeal membrane oxygenation (ECMO) treatment as *ultima ratio* [9,10,11,12].

In developing the Nottingham Physiology Simulator (NPS), Hardman et al. launched an in silico tool for modeling pulmonary disease progression and determining the potential effectiveness of treatment methods [13]. This model was later improved upon by Das et al. and Saffaran et al. to include elements of the cardiovascular system and to improve its performance, which extended its usefulness even further [14,15]. The resulting virtual patient simulator was validated through generating outputs for initial conditions similar to real-world ARDS patients and it was found that these model outputs were consistently comparable with the source clinical data [16]. With a tool such as the NPS, clinicians and biomedical engineers can consistently and accurately model individual patient states, predict the onset of disease and formulate and validate potential treatment methods to guarantee the best outcomes for patients.

The development of models such as the NPS was simplified with the advent of Electronic Health Records (EHRs). Making large amounts of clinical data easily accessible has enabled a lot of research in healthcare, and has helped highlight pathological patterns and uncover treatment methods, but has also sparked discussions about patient privacy and data security [17,18,19]. As these records grow into the realm of Medical Big Data, the need to develop more efficient storage for the data and more capable computing resources to process them grows at a similar rate [20,21,22,23]. Thus, it is essential to make High-Performance Computing (HPC) available for biomedical applications and to develop the algorithms to take advantage of these resources in order to clean, process, analyse and extract information from the available data.

It follows that several teams have already employed available HPC resources in the storage and analysis of Medical Big Data or in training Machine Learning (ML) and Deep Learning (DL) models. Kesselheim et al. applied the Jülich Wizard for European Leadership Science (JUWELS) (https://www.fz-juelich.de/en/ias/jsc/systems/supercomputers/juwels (accessed on 3 February 2023)) supercomputing cluster and booster to perform pre-training of the ResNet-152 DL network. Their goal was to highlight the speed-up achieved using the HPC resources and to eventually perform large-scale transfer learning using the publicly available COVIDx (https://www.kaggle.com/datasets/andyczhao/covidx-cxr2 (accessed on 3 February 2023)) dataset to develop a tool for rapid COVID-19 detection from Chest X-rays (CXRs) [24]. The researchers also discussed using their supercomputing resources to improve the available ML methods for RNA structure prediction. In a similar vein, Baek et al. and Jumper et al. concurrently published their results for the Artificial Intelligence (AI) models RoseTTAFold and AlphaFold, which make use of the HPC clusters available at the University of Washington and at Google, respectively [25,26]. Both teams used an implementation of multi-track DL networks in an attempt to solve the protein-folding problem and in both cases the results were highly accurate. Finally, Zhang et al. made use of HPC to perform hyperparameter tuning on an ML model for Alzheimer’s disease detection [27]. Their work highlights the speed-up that can be achieved by making use of HPC, especially in situations where many trials need to be performed with minute changes in order to find the optimal parameter combination that produces the best results.

This paper describes the process by which an ML and data science platform that takes advantage of Modular Supercomputing Architecture (MSA) available from the Jülich Supercomputing Centre (JSC) is used to build a surrogate model of the NPS with the intention of implementing it for streamlined ARDS-diagnosis support [28,29,30]. In order to achieve this primary goal, several steps need to be completed as follows:Medical data collection, cleaning, analysis and visualisation.Data augmentation through statistical analysis of the available clinical data.Parallel simulation of patient states using a ported NPS.Parallel hyperparameter optimisation of the developed DL model using Ray Tune [31].Final training of the DL-based surrogate model and validation of the results with the original simulation.

As Gherman et al. highlighted, several researchers have already developed ML surrogate models from complex mechanistic models [32]. These surrogates benefit greatly from the high accuracy of the mechanistic models they emulate, while avoiding the computation overhead associated with equilibrating multiple complex differential equations. This aspect coupled with the use of a pre-established HPC-enabled data science and ML platform that was validated in previously published work represent the core innovations of the research described in this manuscript [28,29]. In this way, the HPC resources are instrumental to the accelerated development and testing of the surrogate.

This work is conducted as part of the use case Algorithmic Surveillance of Intensive Care Unit patients with ARDS (ASIC) which is part of the Smart Medical Information Technology for Healthcare (SMITH) project under the guidance of the German Federal Ministry of Education and Research (BMBF) [33,34]. Furthermore, the work described here paves the way for the future development of surrogate models from pre-established mechanistic disease representations, thus providing valuable tools to accelerate diagnosis in critical situations.

## 2. Materials and Methods

The experimental process leading towards completion of the research objective described in the Introduction is represented in Figure 1. The subsections below go further into the details of each step of the experimental process as well as the hardware and software implemented within them.

### 2.1. HPC Resources

The Dynamic Exascale Entry Platform (DEEP) series of projects (https://www.deep-projects.eu/ (accessed on 3 February 2023)) was set up to highlight the benefits of using heterogeneous architectures in HPC to pave the way towards exascale computing by introducing boosters alongside traditional supercomputing clusters [35,36]. The boosters, which run independently of the cluster nodes used for traditional supercomputing tasks, offer the option of expanding storage and compute power for specific tasks, including large-memory nodes for image-processing tasks and multi-GPU nodes for accelerated DL tasks. Thus, the DEEP projects introduced the Modular Supercomputing Architecture (MSA) concept that would later be used in development systems such as the JUWELS cluster and booster, unveiled in 2018 and 2020, respectively [37]. The specific configuration of the cluster-booster prototype set up in the DEEP project and its subsequent projects, DEEP-Extended Reach (DEEP-ER) and DEEP-Extreme Scale Technologies (DEEP-EST), is presented in Table 1.

### 2.2. Software and Libraries

#### 2.2.1. Nottingham Physiology Simulator

The NPS is made available as part of the SMITH project as a central MATLAB (https://www.mathworks.com/products/matlab.html (accessed on 3 February 2023)) script accompanied by peripheral functions written either in MATLAB or in C-script and converted at initial startup into the MATLAB executable (.mex) format. Version 1.4 of the simulator was made available for this research as part of the SMITH project. Further updates to the NPS have already been implemented which improve its performance [15]; however, all of the experiments described in this manuscript concern the version mentioned above. The simulator loads patient data from prepared input files, then runs a preset number of cycles during which it solves a series of differential equations that model the gas exchange occurring during a breathing cycle.

Disease states can be modeled in the simulator through adjusting the input parameters, such as reducing oxygenation, reducing lung compliance or changing the acid–base balance of the blood [16,38], which are typical pathophysiological alterations in ARDS patients [39]. Previous research has validated the performance of the NPS compared to the responses of real patients in the ICU [13,15,40].

Given all of the above, the NPS is certainly a valuable tool in the hands of clinicians aiming to understand medical conditions such as ARDS and to analyse potential treatment methods. It does, however, have specific shortcomings:The time required to run individual simulations makes it unfeasible to use the NPS in diagnosis support, especially for more time-critical clinical situations.The outputs are broad and extremely detailed, requiring users to filter through them in order to extract the information useful for their specific task.It uses proprietary and license-based software, which is a limiting factor for applications on a large scale, especially in remote clinics that would not have proper funding for it.

These shortcomings highlight the need to convert the NPS and to develop the surrogate model as described in the remainder of this manuscript.

#### 2.2.2. Software Used in Model Conversion

As mentioned above, the NPS is built in MATLAB and thus is implemented on a local machine running MATLAB version R2019a within Windows 10 version 22H2. Additionally, the MATLAB Coder (https://www.mathworks.com/products/matlab-coder.html (accessed on 3 February 2023)) software plugin is used in order to export the simulation as a C-script and package it for implementation on the HPC cluster.

The remainder of the programming done for this project uses the Python (https://www.Python.org/ (accessed on 3 February 2023)) programming language with additional packages installed through the built-in pip function or loaded from the list of pre-installed modules available on the HPC cluster. The packages include Numerical Python (NumPy) (https://numpy.org/ (accessed on 3 February 2023)) and Pandas (https://pandas.pydata.org/ (accessed on 3 February 2023)) for data structure manipulation, MatPlotLib (https://matplotlib.org/ (accessed on 3 February 2023)) for data visualisation, Keras (https://keras.io/ (accessed on 3 February 2023)) (running from within TensorFlow (https://www.tensorflow.org/ (accessed on 3 February 2023))) and Scikit-Learn (https://scikit-learn.org/ (accessed on 3 February 2023)) for performing the ML tasks and mpi4py to bind to the Message Passing Interface (MPI) and handle the parallelisation aspect of some of the data-manipulation tasks [41]. Hyperparameter tuning is done using Ray Tune, which in turn employs different scheduling algorithms in order to simplify the task of finding the optimal parameters for training the final model [31]. Finally, the HPC cluster employs the Simple Linux Utility for Resource Management (SLURM) scheduler (https://slurm.schedmd.com/ (accessed on 3 February 2023)) in order to distribute the submitted training and tuning jobs onto the available computing resources. The submission of jobs is done using shell scripts that define the environments to load and the resources to recruit for each specific job.

### 2.3. Model Preparation

In order to build the surrogate model, it is necessary to convert the NPS to a format that can more easily be run in parallel, which would then be used to generate data to train the DL model with. Exporting the model in C-script would be a simple task given its similarity to the MATLAB programming language, as well as the availability of the MATLAB Coder plugin. Accordingly, the various peripheral function files that make up the NPS are grouped into a single script as per the requirements of the MATLAB Coder and the input parameters are defined according to the variables provided in the patient data.

Additionally, the original model outputs an array containing several parameters recorded over every time step of the simulation, which made exporting values difficult. Therefore, the output parameters are reduced to only include the final values of markers for a pulmonary impairment, which can be consistent with an ARDS onset (P_a_O_2_, Partial Pressure of Arterial Carbon Dioxide (P_a_CO_2_), pH and Bicarbonate).

This converted model is tested locally on several patients and its outputs are compared to those from the original simulation in order to verify its integrity. The duration of each simulation is also recorded in order to evaluate the speed-up achieved through this conversion. Moreover, the same patient simulations are performed on the HPC cluster to both validate the outputs and to highlight the speed-up that can be achieved when running several instances concurrently.

### 2.4. Data

The data used in this research were collected from the open-source Medical Information Mart for Intensive Care - III (MIMIC-III) database as part of the research done by Sharafutdinov et al. also within the scope of the SMITH project [42,43,44]. Due to the limited number of patients and the inconsistent representation of their parameters, it was decided to generate simulated data based the statistical distribution of the original data extracted from the MIMIC-III database. In order to perform this data augmentation, the statistical distribution of each parameter listed in Table 2 is analysed and a generator is developed that outputs randomised snapshots of patient states emulating a wide range of real-world parameter combinations. The choice of these parameters was based on the input parameters required for proper functioning of the NPS. Matching the parameters from the simulation to their equivalent values in the MIMIC-III database was done by Sharafutdinov et al. in previous work [43,44]. Table 3 provides a statistical description of the data extracted from the source dataset while Figure 2 presents a comparison of the distributions of the source data and the generated data. In this case, the minimum and maximum cutoff values were chosen based on discussions with clinicians.

As these data are fed into the reduced simulation, the aforementioned markers of ARDS onset of these patients are generated. The end result of this data-manipulation step is a collection of 1,000,000 initial states of patients made up of 19 input parameters and 4 associated expected outputs. The output parameters were chosen based on a sensitivity analysis done by Sharafutdinov et al. in previously published research [44] and are presented in Table 2. The generated patient states are further subdivided into 80% training/10% validation/10% testing datasets to be used to train the DL-based model described in the next section.

### 2.5. Model Design and Training

In order to select the model architecture that offers the greatest potential training performance, several different approaches are tested. However, the choice was limited by two major factors: first, the architecture does not need to be adapted for timeseries data since the inputs chosen are snapshots of patients’ states, as described above; therefore, Recurrent Neural Networks (RNNs) are excluded. Second, no advanced neural network architectures, such as residual layers or transformers, are to be used in order to maintain a reduced model complexity. Accordingly, the models tested out in this step were made up of stacked fully connected layers, convolutional layers or a combination of both.

Several models of both architectures were tested, with varying depths and types of layers, including regularisation, dropout and normalisation layers and with different layer sizes, dropout rates, regularisation factors, learning rates, batch sizes and loss functions. This was done in order to uncover the hyperparameters that have a significant effect on the training process. Each of these architectures was trained for 50 epochs. After this initial testing phase, a provisional best performing model structure is decided on based on a statistical comparison of the four output parameters listed in Table 2 (P_a_O_2_, P_a_CO_2_, pH and HCO_3_) with the outputs generated by the original simulation. Further improvements of this model are done through hyperparameter optimisation as described in the next section.

### 2.6. Hyperparameter Tuning

Hyperparameters are the variables that affect the way in which a model is built or its training process and can be altered either through a process of trial and error or automatically using optimisation algorithms [45,46]. In order to uncover potential hyperparameter combinations through which model training and performance can be improved, the Ray (https://www.ray.io/ (accessed on 3 February 2023)) framework is employed to perform hyperparameter tuning [31,47]. This framework can also take advantage of available HPC resources by distributing the tuning process over several nodes, thus reducing the time needed to run the trials and making the process more efficient.

The schedulers used by Ray Tune in the optimisation process described in this manuscript are HyperBand, Asynchronous HyperBand, Population-Based Training (PBT) and the default First-In, First-Out (FIFO) [48,49,50]. These algorithms distribute the tuning task over the available resources and may interfere with the process by introducing perturbations as is the case for PBT or by shutting down under-performing tasks as is the case for HyperBand and Asynchronous HyperBand. Aside from FIFO, which was chosen to serve as a control in this experiment, the remaining schedulers were chosen based on their purported resource efficiency and accuracy. The comparison of the different algorithms is thus intended to highlight the most successful both in terms of resource use and accuracy of results for this specific application.

In this experiment, the tuned parameters are the learning rate, the batch size, the dropout rate, the loss function and the presence of an intermediate fully connected layer before the output layer in the network architecture. The choice of tuning these specific hyperparameters stemmed from the initial testing carried out in the model design and training phase described in Section 2.5 where changing these parameters had a significant effect on how the models performed. The tuning process is carried out to minimise the validation error value, which serves to reduce the possibility of an overfitting model being selected as the best performing trial. After tuning, the best performing parameters for each scheduler are used to retrain the ML-based model and to highlight the improvement in its prediction performance. Best performance is thus based on the models with the most effective loss reduction and where the output R^2^ scores are closest to 1 for all output parameters. These scores quantify the deviation of the model results from the outputs generated by the original simulation.

## 3. Results

### 3.1. Performance of the C-Based Model

The data generated as per Section 2.4 are used as input for the C-based simulation. To do that, it was necessary to hard code the information into an entry-point function for the simulation, which was done through Python. Additionally and to take advantage of the available HPC resources, the process was automated through a jobscript that recruits the necessary resources and modules, then initialises the aforementioned Python script that in turn scatters the data over the recruited CPUs using MPI. Each worker on the cluster generates its own copies of the entry-point function, compiles and then runs them, then collects the outputs and stores them. When all the tasks are completed successfully, the mother node gathers all the stored outputs, sorts them and then appends them to the original inputs, before saving them as a Comma-Separated Values (CSV) file to be used for training the ML model.

Table 4 presents the average duration of a short (60 min equivalent) and a long (120 min equivalent) simulation in MATLAB and compares it to the average duration of those simulations using the C-based simulation on HPC, which highlights the speed-up that was achieved through this process.

### 3.2. Neural Network Architecture Choice

Different types of neural network architectures with varying depths were tested with the available data. The best performing were based on fully connected layers and on 1D convolutional layers. Further experiments with different depths of the two architectures were performed. From these experiments, it was clear that models built with stacked fully connected layers underperformed compared to the approaches using convolutional layers. Additionally, extending the training duration did not lead to improvement in the results and in some cases led to overfitting.

Applying CNN-based models to the task at hand resulted in more consistent performance even with shallow architectures. Additionally, some of the Convolutional Neural Network (CNN) models did overfit, but in general the models using this architecture reached lower Mean Absolute Error (MAE) in fewer training steps than the fully connected models. The evolution of the MAE during training and validation for several models of both network architectures are presented in Figure 3.

For both of these architectures, several iterations of testing were done during which the depths of the networks were varied, as well as the widths of their layers and the addition of dropout and pooling layers into the network design. This process was done in combination with updating the learning rate, batch size and regularisation rates in order to find a rough estimate of the range of hyperparameters as well as the combination of layers that produced a promising model. Based on the results of these experiments, the parameters to be tuned during the hyperparameter tuning process were selected and the ranges over which the tuning would occur were estimated. Furthermore, the chosen network architecture would be based on CNNs with the possibility of adapting the architecture during the tuning process. Additionally, this network would have four one-dimensional convolution layers, with kernel size of 64 for the first layer and 128 for the remaining layers. The output from the final convolution layer is flattened before being fed either to an intermediate fully connected layer or directly to the output layer.

### 3.3. Hyperparameter-Tuning Results

Four different schedulers were used in the hyperparameter-tuning step of this experiment in order to provide performance comparisons of the different applications. Figure 4 presents the training and validation MAE values of the different trials for each of the schedulers used.

Running on 16 nodes of DEEP-ESB cluster, the FIFO scheduler provides a benchmark as it distributes the 64 available tuning jobs. In this approach, new trials cannot be started until the prescribed maximum number of training epochs of previously scheduled tasks is completed. Completing all the trials required a total of 78 min. The HyperBand scheduler performed early stopping on many trials that were underperforming, which allowed the tuning process to complete within a shorter duration (46 min). Asynchronous HyperBand performed in a similar manner, taking 64 min to complete all the trials. The early stopping is evident in the learning curves of these two scheduling algorithms (Figure 4b,c). Finally, PBT took the longest to complete due to the method with which it implements perturbations at specific times during the model-training process. This is visible in the spikes of the learning curves in Figure 4d. At these points in the training process, the scheduler reruns each trial with a slightly modified learning rate. This resulted in the hyperparameter tuning with the PBT scheduler taking 160 min to complete.

The best performing model parameters from each scheduler are presented in Table 5. A common aspect of the best performing models is the presence of the intermediate fully connected layer before the output layer of the network. Similarly, the dropout rates and learning rates were all within close range for the four models.

### 3.4. Final Model Performance Analysis

The results from each parameter combination listed in Table 5 are presented in Figure 5. The learning curves for the networks trained on the parameters from the FIFO and the Asynchronous HyperBand trials both show some overfitting towards the second half of the training process. This is also reflected in the R^2^ score graphs for the output parameters, where it can be seen that prediction performance for P_a_CO_2_, bicarbonate concentration and pH is reduced compared to the networks from the HyperBand and PBT trials. Additionally, the R^2^ score graphs show that P_a_O_2_ prediction accuracy is consistently lower than the remaining output parameters, although still above 0.90.

## 4. Discussion

Converting the NPS to C helped highlight the speed-up that can be achieved through the use of HPC resources. Running multiple simulations simultaneously as well as the increased efficiency and reduced overhead of C code reduced the code execution times and made it possible to generate more data with which to train the proposed ML models. In the end, the average duration of simulations was less than half the average duration of simulations in MATLAB. Additionally, processing and storing the output data was a computation- and communication-intensive process which was greatly simplified through the availability of online storage on the pre-established HPC-enabled platform for medical ML and data science [30].

The results of the DL model training step of this research highlighted the inherent differences between a fully connected (i.e., traditional Artificial Neural Network (ANN)) architecture and a convolution-based approach. While ANNs are more likely to give value to every input parameter, CNNs are more adapted to uncover connections between the inputs and infer meaning from them, which might explain why these networks consistently performed better. The results in Figure 3 show that CNNs might overfit the data if the layers are not well tuned, but in most cases the performance surpassed that of ANNs and lower MAE values were reached in shorter training periods, which ultimately makes the convolutional approach more resource-efficient. Additionally, the curves consistently show the validation error being lower than the training error; this is due to the regularisation and dropout layers introduced in the network architectures to reduce overfitting. These layers are active during the training process but inactive by design during validation and testing (https://keras.io/getting_started/faq/ (accessed 10 February 2023)).

Similarly, when considering resource efficiency, HyperBand and its successor Asynchronous HyperBand make the best use of the available resources to distribute the available tasks. Besides the reduction in computation time, these two approaches minimise stragglers, that is the number of allocated resources that are not effectively being used for computational tasks. Furthermore, the recommendations from the Ray framework highlight Asynchronous HyperBand as a more capable and efficient scheduler than the original HyperBand (https://docs.ray.io/en/latest/tune/api_docs/schedulers.html (accessed 10 February 2023)). In the case of PBT, resource efficiency is secondary to uncovering more effective approaches through parameter perturbations. Although this approach could be beneficial for applications where minor changes of parameters might greatly alter the outcome of the experiment, the computational overhead necessary for PBT to complete the trials was judged too great for the research purposes described in this manuscript.

The performance of these models is comparable to the performance of the first CNN model in Figure 3, which shows that the best combination of parameters can be reached through a process of trial and error, although it required running several trials to find the best parameters and was extremely time- and resource-consuming and the many combinations were difficult to keep track of. Making use of the hyperparameter-tuning methods streamlined the process and had the added benefit of managing the computation resources and distributing the trials without much interference.

The models trained on the best parameters generated from the tuning process highlight the need to take advantage of early stopping during training. Such an approach might produce better predictive performance from the models trained on the parameters selected by FIFO and Asynchronous HyperBand where overfitting was a clear issue. The model trained on the parameters selected by HyperBand took the longest time to train due to the lower batch size but still had a performance similar to that of the PBT model in terms of R^2^ scores for P_a_CO_2_, bicarbonate concentration and pH. Moreover, it is clear from the results that all models have high prediction accuracy for the four output parameters (R^2^ > 0.90), although the prediction of P_a_O_2_ was consistently lower. This could be due to possible physiological patterns that were not effectively represented within the data, although future tests with larger data sizes might shed more light on the issue.

These results highlight the fact that the surrogate model manages to accurately emulate the performance of the NPS within a statistically acceptable range. Although the performance of the models developed through this approach has not been compared with existing diagnostic support models, the surrogate model benefits greatly from the accuracy that is inherent to the original mechanistic simulation. On the other hand, in replacing the NPS with the DL-based surrogate model, the computational overhead due to nested calculation and equilibration loops is reduced. Additionally, following the experimental procedure described herein, further surrogate models can easily be developed from the NPS with the intent of diagnosing other conditions.

The results described in this research further showcase the benefits of building specialised surrogate models from existing complex medical mechanistic models, a process that is well established in many scientific fields as described by Gherman et al. [32]. Through this process, significantly representative, more easily applicable and more lightweight models can be made available within hospital ICUs. This has the added benefit of not exposing ICUs to unnecessary external threats of data breaches, not requiring specialised and closed-source software and at the same time not exposing the specific inner workings of the models themselves. Furthermore, this approach benefits from the portability of the developed models, as they can be trained within the platform and exported as offline regressors to be implemented within a container environment. These benefits come at the price of slightly reduced accuracy, although the resulting model predictions are still adequate for supporting clinicians in diagnosing potential disease onset and identifying the need for extra medical attention for a given patient. Another shortcoming of the research described herein is the fact that our surrogate is effectively a black box model. This goes against the current *modus operandi* of model development for clinical applications where explainable AI methods are recommended. It follows that developing explainable AI models for clinical diagnosis is one of the research focus points within the developed ML and data science platform described in this manuscript.

## 5. Conclusions

This article described the process by which a pre-established machine learning and data science platform was used to facilitate the conversion of a MATLAB-based virtual patient model. The process took advantage of available HPC infrastructure to parallelise the original model in order to generate synthetic data that was later used to train ML-based surrogate models. The performance of the models was improved through hyperparameter tuning, which also took advantage of parallelisation. The resulting model performance closely mimics the performance of the original model, though with a massive improvement in the speed with which the results are generated. Additionally, the work shed light on the resource use as a means by which to improve efficiency; algorithmic finetuning of the models using parallel computing can efficiently uncover parameter combinations that would otherwise require a long process of trial and error. The work on model conversion is far from complete but offers a glimpse into the clinical applications of virtual patient simulators as real-time diagnostic support tools for clinicians and ICU personnel, especially in situations where early warning can greatly improve outcomes for patients.

## Figures and Tables

**Figure 1 diagnostics-13-02098-f001:**
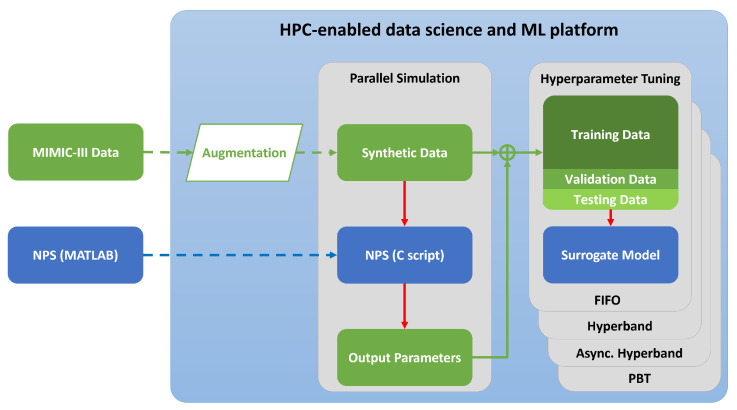
Flow diagram describing the data augmentation and surrogate model development steps within the data analysis and ML platform. The flow of data is represented in green, while the models are represented in blue.

**Figure 2 diagnostics-13-02098-f002:**
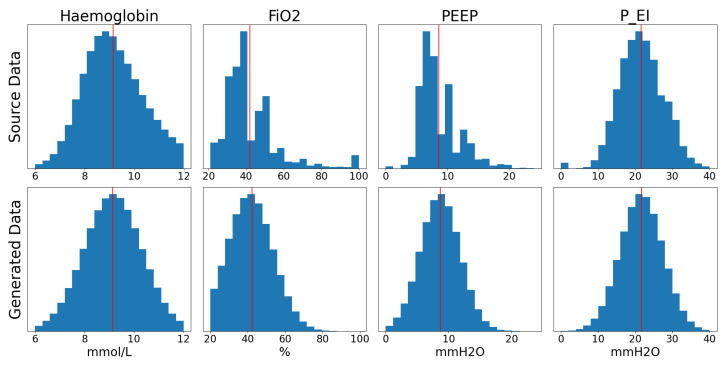
Histograms comparing the distribution of the generated input data with that of the original data. The red lines represent the means for each parameter.

**Figure 3 diagnostics-13-02098-f003:**
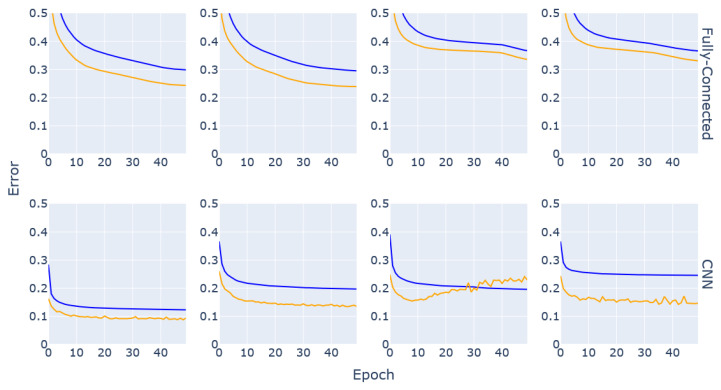
Training (blue) and validation (orange) MAE for several neural networks built either using the fully connected architecture or as CNNs.

**Figure 4 diagnostics-13-02098-f004:**
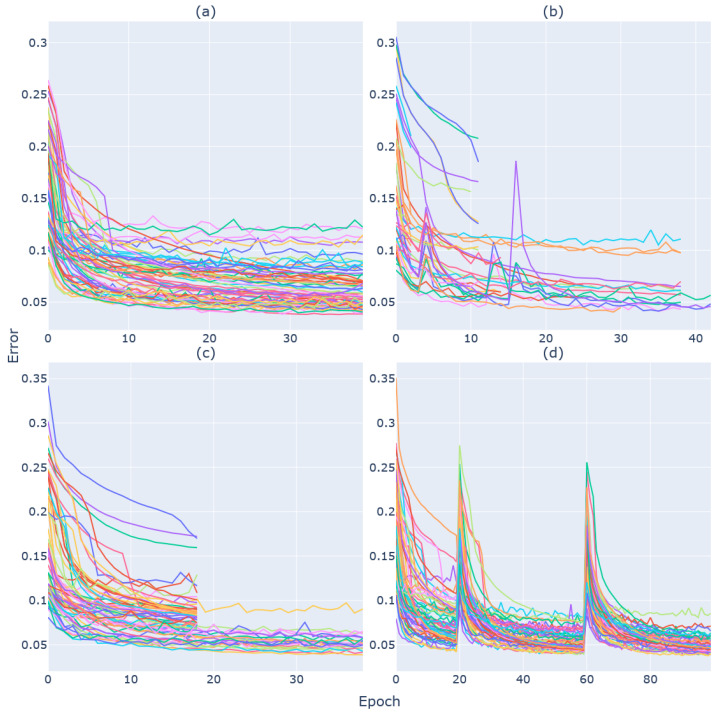
Curves showing the MAE for each of the 64 hyperparameter-tuning trials. Each graph represents the trials for one scheduling algorithm: (**a**) FIFO, (**b**) HyperBand, (**c**) Asynchronous HyperBand and (**d**) PBT.

**Figure 5 diagnostics-13-02098-f005:**
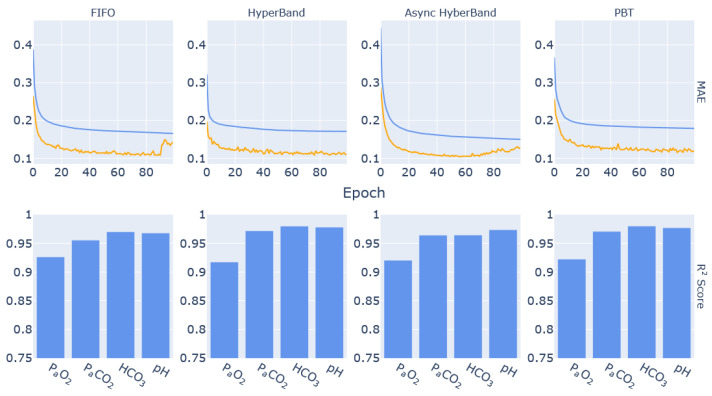
Learning Curves of the Training (blue) and Validation (orange) MAE of the models using the best parameters as selected by each scheduling algorithm, accompanied by their respective per-parameter R^2^ score bar graph.

**Table 1 diagnostics-13-02098-t001:** Partitions on the DEEP Prototype.

Partition	Nodes	CPUs/Node	GPU
DEEP-Data Analytics Module	16	96	NVIDIA V100 + Intel Stratix10 FGPA
DEEP-Extreme Scale Booster	75	16	NVIDIA V100
DEEP-Cluster Module	50	48	n/a

**Table 2 diagnostics-13-02098-t002:** Input and Output parameters of the C-ported virtual patient simulator.

	Parameter	Description
Input Parameters	v_sR, v_inR	Used to calculate individual Compartment Resistance to Flow (R_comp_) values
	v_sVR, v_inVR	Used to calculate individual Compartment Vascular Resistance (VR_comp_) values
	v_nc	Number of Closed Compartments
	asht	Anatomical Shunt
	RQ	Respiratory Quotient
	VO_2_	Oxygen Uptake
	VD_phys_	Volume of Physiological Deadspace
	CO	Cardiac Output
	I:E	Inspiratory to Expiratory Ratio
	Hb	Hæmoglobin
	F_i_O_2_	Fraction of Inspired Oxygen
	PEEP	Peak End-Expiratory Pressure
	P_EI_	End-Inspiratory Pressure
	S_v_O_2_	Venous Oxygen Blood Saturation
	RR	Respiratory Rate
	V_t_	Tidal Volume
	BE_a_	Arterial Base Excess
Output Parameters	P_a_O_2_	Partial Pressure of Arterial Oxygen
	P_a_CO_2_	Partial Pressure of Arterial Carbon Dioxide
	HCO_3_	Bicarbonate Concentration
	pH	Blood Acidity Level

**Table 3 diagnostics-13-02098-t003:** Statistical description of the parameters extracted from the source dataset.

	BE_a_	Hb	V_t_	PEEP	P_EI_	F_i_O_2_	S_v_O_2_	RR
**Unit**	mmol/L	mmol/L	mL	cmH_2_O	cmH_2_O	%	%	
**Min**	−15	6	220	0	0	20	30	10
**Max**	15	12	840	24	40	100	100	40
**Mean**	1.37	9.15	463.80	8.64	21.68	41.08	68.81	20.83
**SD**	4.42	1.17	115.13	3.15	5.76	12.39	11.03	5.73

**Table 4 diagnostics-13-02098-t004:** Comparison of the average duration of the original MATLAB-based simulation with the ported C-code version.

	Short Simulation (run_time = 60)	Long Simulation (run_time = 120)
MATLAB Simulation	51 s	259.1 s
C-based Simulation on HPC	23.1 s	108.8 s

**Table 5 diagnostics-13-02098-t005:** Parameters of the best performing trial from each Hyperparameter-Tuning Scheduler.

Scheduler	Learning Rate	Loss Function	Dropout Rate	Batch Size	Additional Fully Connected Layer
FIFO	$4\times 10^{5}$	MSE	0.5	128	True
HyperBand	$8\times 10^{5}$	MAE	0.52	64	True
Async. HyperBand	$3\times 10^{5}$	MAE	0.5	128	True
PBT	$5.8\times 10^{5}$	MSE	0.54	128	True

## Data Availability

The code for the deep learning models and visualising the outputs as well as the outputs from this experiment are available at https://github.com/c-barakat/ARDS_tune.git accessed on 16 June 2023. The generated patient data is available under https://doi.org/10.23728/B2SHARE.B143C287BB69482A90ABABE7A5A8EB4A accessed on 16 June 2023.

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
