# Peer review of "Developing an Artificial Intelligence-Based Representation of a Virtual Patient Model for Real-Time Diagnosis of Acute Respiratory Distress Syndrome"

_diagnostics, 2023, doi:10.3390/diagnostics13122098_

Round 1
Reviewer 1 Report
The authors have presented the development of a deep-learning model to simulate the onset of acute respiratory distress syndrome in a virtual patient. The process is explained starting from the initial parameters to the creation of simulated data, and the creation and training of the model.
I suggest to provide a more detailed description of the simulated dataset, including descriptive statistics.
I also suggest improving the description of the model output and how it was compared with the simulator.
Author Response
Thank you very much for your feedback. We hope the corrections made and listed in the attached document properly address your comments.
Please see attachment

Reviewer 2 Report
The article presents development of DL surrogate model on the ARDS dataset for virtual patient. I have certain comments for the improvement of the article.
1. Abstract- Clearly mention the specific contributions of the paper. For example, you can highlight the novelty of developing a deep learning-based surrogate model for ARDS onset prediction using the Nottingham Physiology Simulator. Explain how this approach differs from existing methods and how it addresses the limitations of previous models.
2. Introduction- (a) Clearly state the objectives or research questions of the study at the end of the introduction. This will help readers understand the specific goals the paper aims to achieve.
(b) Expand on the significance and impact of ARDS. Explain the mortality rate, the challenges in diagnosing the condition, and the need for effective treatment methods.
(c) Clearly explain the purpose and significance of the NPS as a tool for modeling pulmonary disease progression and evaluating treatment methods.
3. Related Work- More recent papers needs to be added.
4. No problem formulation is done. No algorithm or flow diagram is presented. What are the conditions which are to be optimized? Under what constraints
5. More comparitive results needs to be added with previous works.
Moderate changes is required.
Author Response
Thank you very much for your feedback on our article. We hope the corrections listed in the attached document provide an adequate response to your comments.
Additionally, we proofread the manuscript to make sure that no grammatical and/or spelling mistakes have made their way into the final version.
Please see the attachment.
